# DNA barcoding indicates multiple invasions of the freshwater snail *Melanoides tuberculata sensu lato* in Florida

**Lori R. Tolley-Jordan[1], Michael A. Chadwick[2]\*, Jimmy K. Triplett[1]**

**1** Dept. of Biology, Jacksonville State University, Jacksonville, AL, United States of America, **2** Dept. of Geography, King's College London, London, United Kingdom

\* michael.chadwick@kcl.ac.uk

## Abstract

*Melanoides tuberculata sensu lato* (Thiaridae) are polymorphic female-clonal snails of Asian and African origins that have invaded freshwaters worldwide, including those in Florida. Although the snails have been documented in Florida for at least 70 years, no studies have investigated whether the observed distribution is due to a single introduction or multiple independent invasions. Here, cytochrome oxidase I was used to measure genetic diversity within and among sites in Florida and compare genetic diversity between Florida and other regions of the world. We also examined the relationship between shell morphology and haplotype diversity to determine if shell morphs can serve as a proxy for haplotypes. In total, we recovered 8 haplotypes randomly distributed across populations in Florida. Phylogenetic reconstruction supported the hypothesis of multiple invasions by diverse representatives of the *M. tuberculata* species complex. In contrast, shell morphology was not found to be a useful phylogeographic character, with divergent haplotypes represented by similar shell forms. These results suggest that the observed invasion patterns in Florida are best explained by serial introductions, and that shell morphology cannot be used to predict haplotypes or reconstruct invasion history of *Melanoides tuberculata s.l.* and that extensive taxonomic revisions are needed to investigate invasion dynamics.

## Introduction

*Melanoides tuberculata* (Müller 1774) is a freshwater snail in the family Thiaridae, endemic to Africa and Asia. Recent findings by Wiggering [1] showed that most Thiarid genera and species are polyphyletic, including *M. tuberculata*. Thus, we recognize this species in the broad sense (*M. tuberculata sensu lato*), and all subsequent references assume this taxonomic interpretation. In the past century, *M. tuberculata* has successfully invaded tropical/sub-tropical waters and temperate warm-water springs globally (see review in [2]). The snails can survive a wide range of environmental stressors [3] including desiccation [4], salinity [5], and high temperatures [6]. Moreover, parthenogenetic iteroparous reproduction and generalist feeding strategies [3, 7] allow populations to reach densities of >100 individuals per m$^2$ [8–10].

**Data Availability Statement:** Relevant data are within the paper. Additionally, all sequence data have been uploaded to GenBank.

**Funding:** Funding was provided by United States Fish and Wildlife Service, Gulf States Marine Fisheries Division (project ID: FWS-800-037-2016-JSU) and the McCluer Scholar Award given to L. Tolley-Jordan by the School of Science, Jacksonville State University. The funders had no role in study design, data collection and analysis, decision to publish, or preparation of the manuscript.

**Competing interests:** The authors have declared that no competing interests exist.

Unfortunately, due to their ubiquitous presence in the aquarium trade [11–13], human-mediated spread is likely the main invasion pathway [11, 14–16], although specific introduction mechanisms (e.g. intentional or unintentional escapes) are poorly reported and difficult to detect [17].

Across the USA, Florida has the greatest number of established invasive aquatic species [18]. In particular, *M. tuberculata* is widespread in springs, canals, and wetlands throughout the state [6, 14, 19, 20]. The snails were first discovered in Florida in the mid-1960s at multiple locations over a 5-year period (Table 1). These records suggest that the snails invaded years earlier and spread throughout the state, or else multiple simultaneous introductions occurred. The current widespread distribution in Florida is not surprising given the ideal environmental conditions combined with biological traits that favor individual survival and reproduction, the prolific occurrence of *M. tuberculata* in the aquarium trade, the inter-connectedness of natural hydrologic features such as springs and wetlands [21], and anthropogenic feature like canals and flood defences [22].

Native populations of *M. tuberculata* in Southeast Asia, the Middle East, India, and East Africa have distinct haplotypes that can be used in a phylogenetic context to infer the region of origin of snail invasions and subsequent dispersions [1, 23, 24]. High haplotype diversity in an invaded region has been suggested to indicate multiple introductions [25, 26], while low genetic structuring (i.e., few haplotypes) is more likely to indicate limited introductions followed by range expansion [23, 27].

As invasive populations of *M. tuberculata* are primarily females that reproduce via parthenogenesis, haplotypes are generally conserved [28] and relatively easy to match to putative source populations via barcoding with mitochondrial (cytochrome oxidase I, COI) and ribosomal markers (16s rDNA) [7, 11, 16, 24, 29, 30]. Additionally, studies have shown that unique haplotypes are represented by unique, conserved shell morphs [7, 24, 30, 31], providing a useful tool for resource managers to track invasions. For example, unique shell morphs linked to distinct haplotypes have been used to verify that invasive lineages in New World waterways and East African lakes correspond to Southeast Asian populations [7, 24]. However, shell morph analyses can be difficult to interpret due to potential phenotypic plasticity [7, 32–34] and hybridization [31]. Regardless, the combination of genetic and morphological methods

**Table 1. Earliest locality records from 1966 to 1972 of *M. tuberculata* from waterbodies throughout Florida.** Records were queried from databases held by: Florida Museum of Natural History (FLMNH), North Carolina State Museum (NSCM), and the U.S. Geological Survey Nonindigenous Aquatic Species Database (ANS - https://nas.er.usgs.gov/). ND = no data.

| Locality | Latitude | Longitude | Year | Collector | Database |
|---|---|---|---|---|---|
| Lake Osceola, Miami | 25.7183 | -80.2793 | 1966 | Clench, W. | FLMNH |
| Pond, Miami | 25.9420 | -80.1553 | 1968 | Clench, W. | FLMNH |
| Lithia Springs, Tampa | 27.8663 | -82.2308 | 1969 | ND | FLMNH |
| Hillsborough River, Tampa | 27.9668 | -82.4747 | 1969 | ND | FLMNH |
| Hillsborough River, Tampa | 28.1039 | -82.2781 | 1969 | Clench, W. | FLMNH |
| Purity Springs, Tampa | 28.0219 | -82.4620 | 1969 | Thompson, F. | ANS |
| Ponds, Melbourne | 28.0836 | -80.6081 | 1970 | Thompson, F. | FLMNH |
| Rainbow Springs | 29.0697 | -82.4272 | 1970 | Athearn, H. | NCSM |
| Middle River Canal, Ft. Lauderdale | 26.1727 | -80.2289 | 1971 | Russo, T. | FLMNH |
| Pompano Canal, Ft. Lauderdale | 26.2315 | -80.1492 | 1971 | Russo, T. | FLMNH |
| Gulf Stream, Ft. Lauderdale | 26.3155 | -80.0825 | 1971 | Reinert, J. | ANS |
| Miami R., Miami | 25.7793 | -80.2087 | 1971 | Russo, T. | ANS |
| Fish Eating Creek, Lakeport | 26.9309 | -81.2881 | 1971 | Athearn, H. | NCSM |
| Lake Okeechobee, Lakeport | 26.8236 | -80.6706 | 1972 | Athearn, H. | NCSM |

can provide evidence for evaluating putative sources of introductions and subsequent spread of these invasive snails.

Although definitive transport pathways of invasive snails to Florida's waterways cannot be determined, available sequences of these snails on GenBank with accompanying locality data can be used to compare population genetic structure with other studies and infer invasion dynamics. We used a barcoding gene (mitochondrial cytochrome oxidase I, COI) to characterized Florida snail haplotypes and compared these to global haplotypes in a phylogeographic framework. This allowed us to evaluate two invasion scenarios which may drive local biogeography: (1) serial founder events from multiple, putative source populations, and (2) limited introductions and subsequent range expansions. In addition, we explored the fidelity of shell forms to haplotypes to evaluate if rapid, inexpensive morphometric methods can be used to detect invasion origins independently of genetic analyses.

## Methods

### Snail collections

We surveyed 25 waterbodies throughout Florida [35]. From each site where snails were found, we collected approximately 10 individuals for molecular analyses. From all individuals, head tissues were removed and preserved in 95% molecular grade ethanol. In addition, tissues from 4 individuals of *Tarebia granifera* collected from Florida were included in the analyses. All preserved snails were vouchered at the Jacksonville State University invertebrate collection.

### DNA extraction, amplification, and sequencing

DNA was extracted and isolated from head tissue using 20μL proteinase K with 180μL ATL buffer from a Qiagen DNeasy Blood and Tissue Kit and the process was completed using a Qiagen Plant Extraction Kit. We used a combined method for DNA extraction because polysaccharides found in the snail tissue act as a polymerase chain reaction (PCR) inhibitor [36]. A fragment of the mitochondrial cytochrome oxidase I (COI) gene was amplified using the forward primer LCO1490 (5′-GGTCAACAAATCATAAAGATATTGG-3′) and the reverse primer HCO2198 (5′-TAAACTTCAGGGTGACCAAAAAATCA-3′; [37]) in PCR carried out in 25μL volumes using 12.5μL of BioMix Red (Bioline, Taunton, MA), 2.5μL (10 mM) of the forward and reverse primers described above, 5μL of sterile deionized water, and 5μL of DNA. PCR was performed in a thermocycler using a cycling profile of initial denaturation at 95˚C for one minute and 35 cycles of denaturation at 95˚C for one minute, annealing at 56˚C for one minute, and elongation at 72˚C for one and a half minutes, followed by a final elongation step at 72˚C for seven minutes. Amplicons were confirmed on a 1% agarose gel stained with SYBR Green (ThermoFisher Scientific). Amplicons were cleaned using ExoSAPIT (Affymetrix) following the manufacturer's protocol. DNA sequencing was completed at Molecular Cloning Laboratories (MCLAB; South San Francisco, CA). All individual sequences from this study are available in NCBI: GenBank (*Tarebia granifera*: MT671805—MT671808 and *M. tuberculata*: OP114776—OP11824).

### Molecular analyses: Haplotype diversity and phylogeography

COI amplicons ranging from 570 to 641 bp from 49 individuals and 11 sampled locations were assembled, verified, and aligned using the MUSCLE algorithm in MEGA X [38]. These are subsequently referred to as the 'FLORIDA' dataset. Haplotype network analysis for our FLORIDA dataset was conducted in RStudio in the PEGAS package using the haplotype and haploNet functions [39, 40]. FLORIDA haplotypes were then aligned with COI sequences from

**Table 2. Accessioned sequences of Thiaridae and Paludomidae from NCBI: GenBank included in phylogenetic analyses (n = 60).** Species names, locations, and invasion status (N = native and I = Invaded) are from GenBank data. N/A are sequences in which the origin is not listed by the author. More detailed localities are provided in Fig 3.

| Species | GenBank | # of sequences | Location |
|---|---|---|---|
| *Melanoides imitatrix* | DQ995480 | 1 | Africa, N |
| *Melanoides polymorpha* | AY958728-958753 | 3 | Africa, N |
| *Melanoides sp.* | AY213150 | 1 | Africa, N |
| *Melanoides tuberculata* | AF236071 | 1 | French Polynesia, I |
| *Melanoides tuberculata* | AY456563-456564 | 2 | Africa, N |
| *Melanoides tuberculata* | AY575971-575995 | 14 | Africa, N & I |
| | | | Middle East, N |
| | | | Southeast Asia, N |
| *Melanoides tuberculata* | KP284130-284139 | 10 | Southeast Asia, N |
| *Melanoides tuberculata* | KP774674-774711 | 7 | Southeast Asia, N |
| *Melanoides tuberculata* | KT280408-280412 | 3 | Middle East, NA |
| *Melanoides tuberculata* | MK697710-697712 | 3 | Southeast Asia, N |
| *Melanoides tuberculata* | MK697735 | 1 | India, N |
| *Melanoides tuberculata* | MK879274 | 1 | India, N |
| *Melanoides tuberculata* | MT499035-499099 | 4 | Australia, I |
| *Paludomus siamensis* | MK879286 | 1 | Southeast Asia, N |
| *Stenomelania aspirans* | MK697723 | 1 | Indonesia, N |
| *Stenomelania crenulata* | AB920321 | 1 | Southeast Asia, N |
| *Stenomelania denisoniensis* | MK697732 | 1 | Southeast Asia, N |
| *Stenomelania offachiensis* | KU318343 | 1 | Southeast Asia, N |
| *Stenomelania sp.* | MH319880 | 1 | Southeast Asia, N |
| *Tarebia granifera* | MK000291-000353 | 2 | Southeast Asia, N |
| *Tarebia granifera* | MT671805 | 1 | USA, I |

Thiaridae snail species available on GenBank (subsequently referred to as 'WORLD'; Table 2) in MEGA X [38]. Comparisons of the genetic diversity including number of polymorphic sites, haplotype diversity (hd), nucleotide diversity (π), and shared haplotypes between FLORIDA and the WORLD were calculated in DnaSP 6.12.03 [41]. Genetic structure was determined by an analysis of molecular variance (AMOVA [42]) using the software ARLEQUIN version 3.5 with significance of variance components tested using 10,000 nonparametric permutations of individuals among groups. We also used ARLEQUIN to calculate pairwise F-Statistics by grouping all haplotype sequences into four metapopulations: FLORIDA and three groups from WORLD: 1) all sequences other than FLORIDA, 2) sequences from individuals collected in Africa, India, and the Middle East that were considered native (AIM-Native), 3) sequences from individuals collected in Africa, Australia, India, the Middle East, and French Polynesia that were considered invasive in other studies (AIM-Invasive), and 4) sequences from individuals collected in India and Southeast Asia (ASIA) that were considered native in other studies (see Table 2).

Phylogenetic reconstruction was used to assess putative geographic origins of the FLORIDA sequences via comparison with the GenBank WORLD dataset [41]. The 49 sequences from FLORIDA were collapsed into unique haplotypes and aligned with 46 WORLD sequences along with an additional 13 sequences of other Thiaridae spp. including *Melanoides spp.*, *Stenomelania spp.*, and *Tarebia granifera* (Table 2). *Paludomus siamensis* was used as the outgroup for phylogenetic analyses; and included in the 70 aligned sequences of 586 characters were analysed with maximum parsimony (MP) in PAUP* [43]. Node stability was determined

by bootstrap resampling of 1000 replicates of heuristic searches. The data were also analysed using Bayesian inference (BI) in MrBayes 3.2 [44]. BI analyses were conducted using a partitioned GTR + I + G model for reasons outlined by Huelsenbeck and Rannala [45], with all parameter values estimated during the analysis. A Dirichlet prior was used for base frequencies and the rate matrix. A uniform prior was used for the shape parameter ($\alpha$), proportion of invariable sites (I), and topology. Branch lengths were unconstrained. Four separate MCMC runs were initiated, each with 10,000,000 generations. Runs were started from a random tree; the topology was sampled every 1,000 generations of the MCMC chain. Majority rule (50%) consensus trees were constructed after removing the first 10% of sampled trees ("burn-in"). Branch support was assessed according to a 0.95 posterior probability measure for BI [46, 47].

## Shell morphology

Prior to evaluating morphological features on shells, specimens were cleaned with soap and water and a soft bristle brush to remove deposits. Cleaning was done delicately so as not to damage any shell features. A scoring system developed by Facon et al. [24] and modified by Genner et al. [48] and Van Bocxlaer et al. [7] was used to evaluate a range of shell characteristics. All scores listed as not applicable were assigned a score of zero in the analyses. Shells with the same exact set of scores were grouped and this matrix of shell characteristic was then evaluated using non-dimensional scaling analyses (NDS) in PAST v.4.03 [49]. NDS results were used to identify discrete shell morph groups and then link these to corresponding haplotypes (see above).

## Results

From the 25 locations sampled, 11 were found to have live snails (Table 1, [34]) and 350 individuals were collected. At four sites, < 10 snails for genetic analyses were collected (sites 1, 6, and 24 = 6 individuals; site 3 = 2 individuals). At these sites, only individuals < 8 mm were collected, and these individuals were not included in morphological analyses.

### Genetic structuring and lineage origins

In total, 8 haplotypes were recovered from snails collected in Florida (Fig 1). Haplotype *c* (representing 40% of Florida haplotypes) was the most widely distributed, being found at five sites on both east (sites 3 and 4) and west coasts (site 19) and in southern Florida (sites 13 and 15 (Fig 1). Haplotype *d* (representing 27% of Florida haplotypes) was found at three sites ranging from north-eastern (site 25) and coastal central Florida (sites 1 and 24; Fig 1). Haplotype *h* occurred at two sites on the central east coast (19 and 24). Two sites had two haplotypes (site 13: haplotypes *b*, *c*; site 15: haplotypes *a*, *c*) and 1 site had 3 haplotypes (site 19: haplotypes *b*, *f*, *g*; Fig 1). All other sites were represented by a single haplotype (Fig 1). We found low shell morph fidelity to unique haplotypes. For example, shell morphs FLAA and FLAD, which were close together in morpho-space (Fig 2), represented haplotypes in two different clades (Fig 1). Conversely, haplotypes *a*, *c*, and *d* (clade 1) and haplotypes *g*, *h* (clade 2) converged on different shell morphs (FLAB and FLAA, respectfully). Additionally, we found different shell morphs for the same haplotype (i.e., haplotypes *a* and *g*) from different sampling locations. As such, haplotypes in Florida display a strong degree of phenotypic plasticity.

In total, 30 haplotypes were identified when FLORIDA and WORLD sequences were combined. Of the 8 haplotypes recovered from FLORIDA, 7 were shared with haplotypes from WORLD (Table 3). The AMOVA revealed no significant differences in genetic structure between FLORIDA and WORLD, as most of the variation occurred within metapopulations

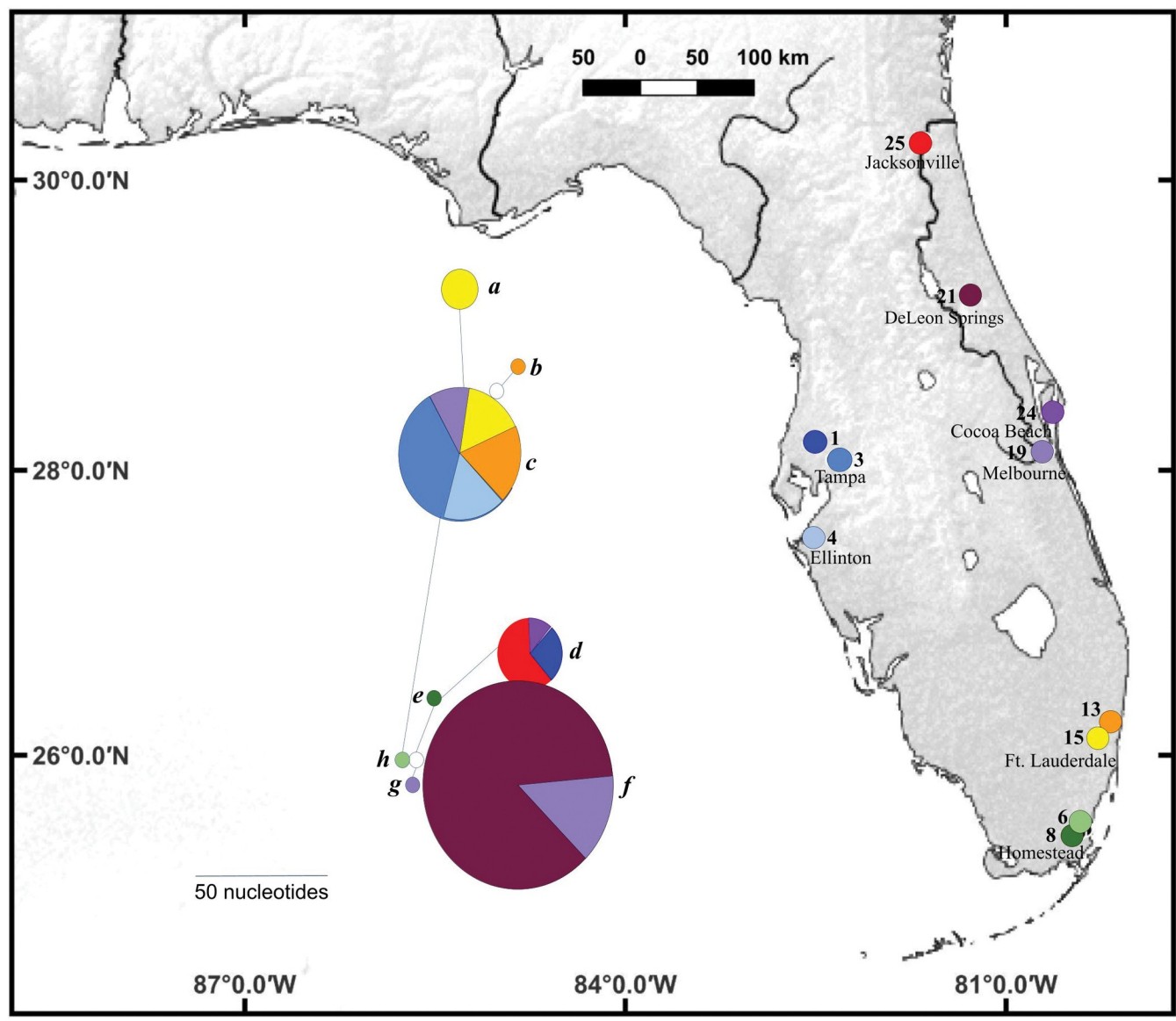

**Fig 1. CO1 haplotype network of putative *Melanoides tuberculata* from 11 sites.** Circle sizes reflect haplotype frequency. Open circles represent 1 nucleotide difference from nearest haplotype and are not considered unique based on DnaSP haplotype determinations. Italicized letters a-h are haplotype identifiers and colors in network correspond to sites where individuals were collected. The map was made using QGIS 3.2) using a public domain map dataset from www.naturalearth.com.

(94.54%; Table 4), and a small amount (5.92%) was explained by variation among metapopulations; no variation was detected between groups (FLORIDA versus WORLD, Table 4).

The average number of pairwise differences within populations ranged from 0.84559 (FLORIDA) to 0.97386 (ASIA), consistent with the high level of variation attributed to metapopulations. Pairwise $F_{ST}$ values (Table 5) indicated little to moderate genetic differentiation between metapopulations (0–0.10; [50]), with the highest levels of genetic differentiation between Florida and AIM-Native ($F_{ST}$ = 0.09842). The mean number of pairwise differences between metapopulations ranged from 0.90196 between FLORIDA and AIM-Invasive, and 1.00000 between AIM-Native and AIM-Invasive (Table 5).

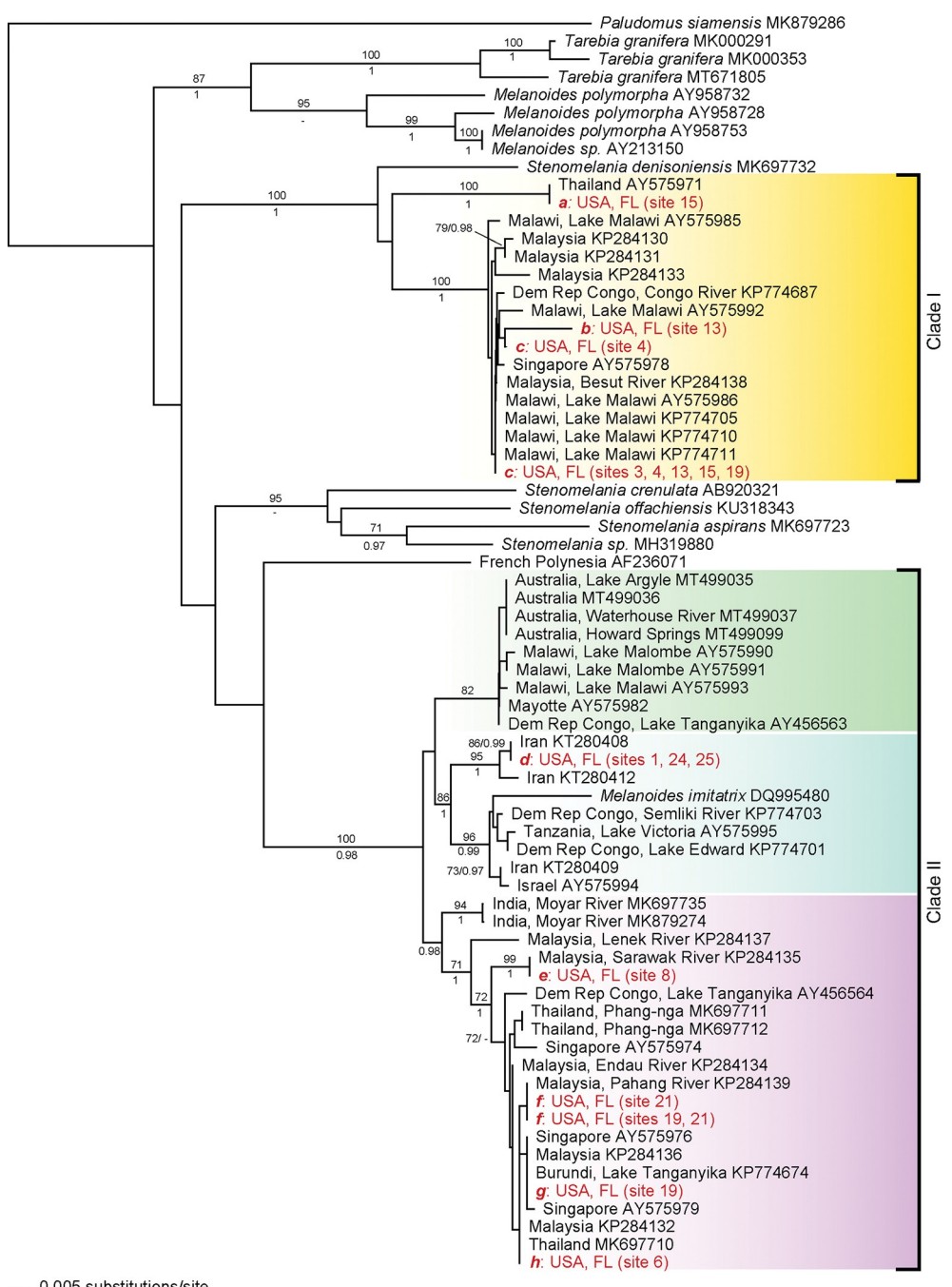

**Fig 2. 50% Majority consensus tree based on COI sequences.** Numbers at nodes show bootstrap values from Parsimony (top) and posterior probabilities values from Bayesian (bottom) analyses. Parsimony analyses resulted 185 parsimony informative characters in 126 retained topologies with a consistency index of 0.5118 and a retention index of 0.8811. The two recovered clades including FLORIDA and WORLD sequences are shown with vertical bars. Clade I is represented by gold shading and Clade II includes shading for each subclade with green (subclade 1), blue (subclade 2), and purple (subclade 3).

Phylogenetic reconstruction resulted in a well-supported tree that revealed both *Stenomelania spp.* and *Melanoides spp.* to be polyphyletic. All WORLD and FLORIDA sequences, except French Polynesia, were separated into two well-supported, disjunct clades (Fig 2). The

**Table 3. Genetic variation of COI data from the total dataset, WORLD, and FLORIDA sequences.** The numbers of polymorphic sites, haplotype numbers, shared haplotypes; haplotype diversity ($H_d$ ± standard error) and nucleotide diversity ($\pi$ ± standard error) are reported for each category; n = number of sequences.

| Population | n | polymorphic sites | haplotypes | $H_d$ ± SD | $\pi$ ± SD |
|---|---|---|---|---|---|
| Total | 95 | 139 | 30 | 0.896 ± 0.021 | 0.077 ± 0.002 |
| WORLD | 46 | 133 | 29 | 0.972 ± 0.011 | 0.073 ± 0.006 |
| FLORIDA | 49 | 100 | 8 | 0.743 ± 0.040 | 0.077 ± 0.005 |

sequence from French Polynesia (AIM-Invasive) was sister to clade 2 and was not included as a member of the clade. Clade 1 included 3 haplotypes from FLORIDA, 5 haplotypes from ASIA, and 9 haplotypes from AIM-Invasive. FLORIDA haplotype *a* shared a haplotype with individuals native to Thailand (ASIA), *c* shared a haplotype with individuals native to Singapore (ASIA) and East Africa (AIM-Invasive). Haplotype *b* was unique to FLORIDA but embedded in the clade with haplotype *c*. Clade 2 was further resolved into 4 subclades, of which two included sequences from FLORIDA. Subclade 1 included AIM-Invasive sequences from Africa and Australia, Subclade 2 included AIM-Native sequences and one FLORIDA haplotype (*d*) that shared a sequence with individuals from Iran (AIM-Native), and Subclade 3 contained sequences from India (AIM-Native), East Africa (AIM-Invasive), Malaysia, Singapore, and Thailand (ASIA) and FLORIDA. Haplotypes *e*, *f*, and *h* each shared a haplotype with sequences from ASIA, while haplotype *g* shared a haplotype both with ASIA and East Africa (AIM-invasive).

## Shell morph fidelity to haplotypes

Shell character analyses resulted in seven distinct morphs (NDS stress = 0.06; Fig 2, Table 6). Both clades had unique shell morphs (Fig 3.; i.e., no shell morph occurred in both clades 1 and 2). In general, haplotypes were found to be phenotypically plastic, since a single haplotype could be represented by more than one morph (e.g. haplotype *c* and morphs FLAA and FLAC; haplotype *d* and morphs FLAA, FLAC, and FLAD). However, in some cases a single shell morph was represented by more than one haplotype (for example, haplotypes *b*, *c*, and *d* were all represented by morph FLAA, and haplotypes *f*, *h*, and *i* were represented by morph FLAB). Only haplotypes *a* and *e* were represented by single shell morphs (FLAE and FLAG, respectively).

## Discussion

*Melanoides tuberculata sensu lato* is a taxonomically complex polyphyletic group, best characterized as a cryptic species complex [1, 34] and in need of taxonomic revision [51]. Despite this taxonomic uncertainty, the current study demonstrates the value of a phylogenetic analysis of molecular data to resolve the history of invasive species (however see [52]). The inclusion of 49 sequenced individuals from this study along with a geographically diverse sample of GenBank sequences, and the outgroup *Paludomus siamensis*, Paludomidae, the presumed sister lineage

**Table 4. Analysis of Molecular Variance (AMOVA) of COI data comparing genetic structure between groups (FLORIDA and WORLD) and among and within the designated metapopulations.**

| Source of variation | df | Sum of squares | Variance components | % of variation | P |
|---|---|---|---|---|---|
| Among groups | 1 | 0.87 | 0.874 | 0 | 0.50 |
| Among metapopulations | 2 | 1.78 | 0.002 | 5.92 | < 0.0001 |
| Within metapopulations | 59 | 28.58 | 0.285 | 94.54 | < 0.0001 |
| Total | 62 | 29.51 | 0.482 | | |

**Table 5. Genetic distances among the designated metapopulations.** The average number of pairwise differences within populations are indicated on the diagonal (bold). Pairwise $F_{ST}$ indices are below diagonal and average number of pairwise differences are above diagonal. Significance of $F_{ST}$ indices tested by a permutation analysis against alternative random partitioning of individuals (10 000 permutations). Asterisks indicate comparisons $P < 0.05$.

|  | FLORIDA | AIM-Native | AIM-Invasive | ASIA |
|---|---|---|---|---|
| **FLORIDA** | **0.84559** | 0.98643 | 0.90196 | 0.94771 |
| **AIM-Native** | 0.09842* | **0.93590** | 1.00000 | 0.99573 |
| **AIM-INV** | 0.04049 | 0.08961* | **0.88571** | 0.97407 |
| **ASIA** | 0.03979 | 0.04049* | 0.04478* | **0.97386** |

to the monophyletic Thiaridae [51] ovided for a robust phylogenetic dataset to evaluate historical snail invasion patterns of *M. tuberculata sensu lato* in Florida (Figs 1 and 2). Like other studies, we found molecular evidence supports the hypothesis of multiple introductions from a range of potential source populations, based on the distribution of FLORIDA sequences in both clades (Fig 2) of this polyphyletic species complex [7, 16, 24, 30, 48]. In contrast, shell morphology was not found to be a useful tool to evaluate invasion dynamics (Fig 3). Regardless, our results provide needed evidence to evaluate likely scenarios driving snail invasion

**Table 6. Summary of FLORIDA shell morph characteristics.** Shell colors, patterns, columella patterns, shape, and sculpture characters described in Facon et al. [24].

| Morphs | background color | | | color patterns | | | | columellar band | | general shape | | sculpture | | |
|---|---|---|---|---|---|---|---|---|---|---|---|---|---|---|
|  | IN | TI | HE | DO | SP | SO | HO | SH | SC | CO | RO | GR | RI | RD |
| FLAA | 2 | 2 | 0 | 2 | 3 | 1 | 2 | 3 | 2 | 2 | 2 | 2 | 3 | 3 |
| FLAB | 3 | 2 | 0 | 2 | 2 | 3 | 1 | 3 | 3 | 1 | 2 | 1 | 0 | 1 |
| FLAC | 4 | 3 | 0 | 0 | 0 | 0 | 0 | 0 | 0 | 2 | 3 | 3 | 3 | 3 |
| FLAD | 2 | 1 | 0 | 2 | 2 | 2 | 3 | 1 | 0 | 1 | 1 | 1 | 0 | 0 |
| FLAE | 3 | 1 | 0 | 0 | 0 | 0 | 0 | 2 | 2 | 1 | 1 | 0 | 0 | 0 |
| FLAF | 2 | 2 | 1 | 2 | 2 | 3 | 1 | 1 | 0 | 2 | 3 | 2 | 3 | 2 |
| FLAG | 2 | 2 | 1 | 2 | 2 | 2 | 2 | 2 | 2 | 2 | 2 | 2 | 2 | 2 |

Character Explanation

Background color

IN-intensity of the shell background color: (1) very pale (2) pale (3) medium (4) dark

TI-background tint of the shell: (1) yellow to brown (2) greenish (3) orange to reddish (4) white (5) black

HE-heterogeneity of the background color among different parts of a shell whorl: (0) homogenous (1) distinctly darker band below the suture

Color pattern

DO-overall density of reddish-brown color patterning on the whole shell, except one just below sutures: (0) no patterning (1) medium (2) dense

SP-type of patterning, expressed as the proportion of spots vs. flames (0) only flames (1) more flames than spots (2) more spots than flames (3) only spots

SO-size of individual spots/flames: (1) small spots/marrow flames (2) medium (3) large spots/wide flames

HO–heterogeneity of color patterning among different parts of the whorl: (1) homogenous (2) slightly different patterns just below sutures (3) strongly different patterns in the subsutural zone compared to those on the rest of the shell

Columellar band

SH-size of the columellar band, when present: (1) narrow, (2) medium, (3) wide.

SC-size of the dark band when present: (1) narrow (2) medium (3) wide

General shape

CO-conicity of the shell: (1) acute (2) medium (3) blunted cone

RO-roundness of the body whorl: (1) flat (2) slightly rounded (3) well-rounded

Sculpture

GR-spiral cords/grooves: (0) absent (1) shallow grooves/weakly pronounced cords (2) intermediate (3) very deep grooves/strongly pronounced cords

Ri-density and width of axial ribs: (0) none (1)a few narrow ribs (2) a few large ribs (3) many narrow ribs

RD-depth of axial ribs when present (1) shallow (2) medium (3) deep

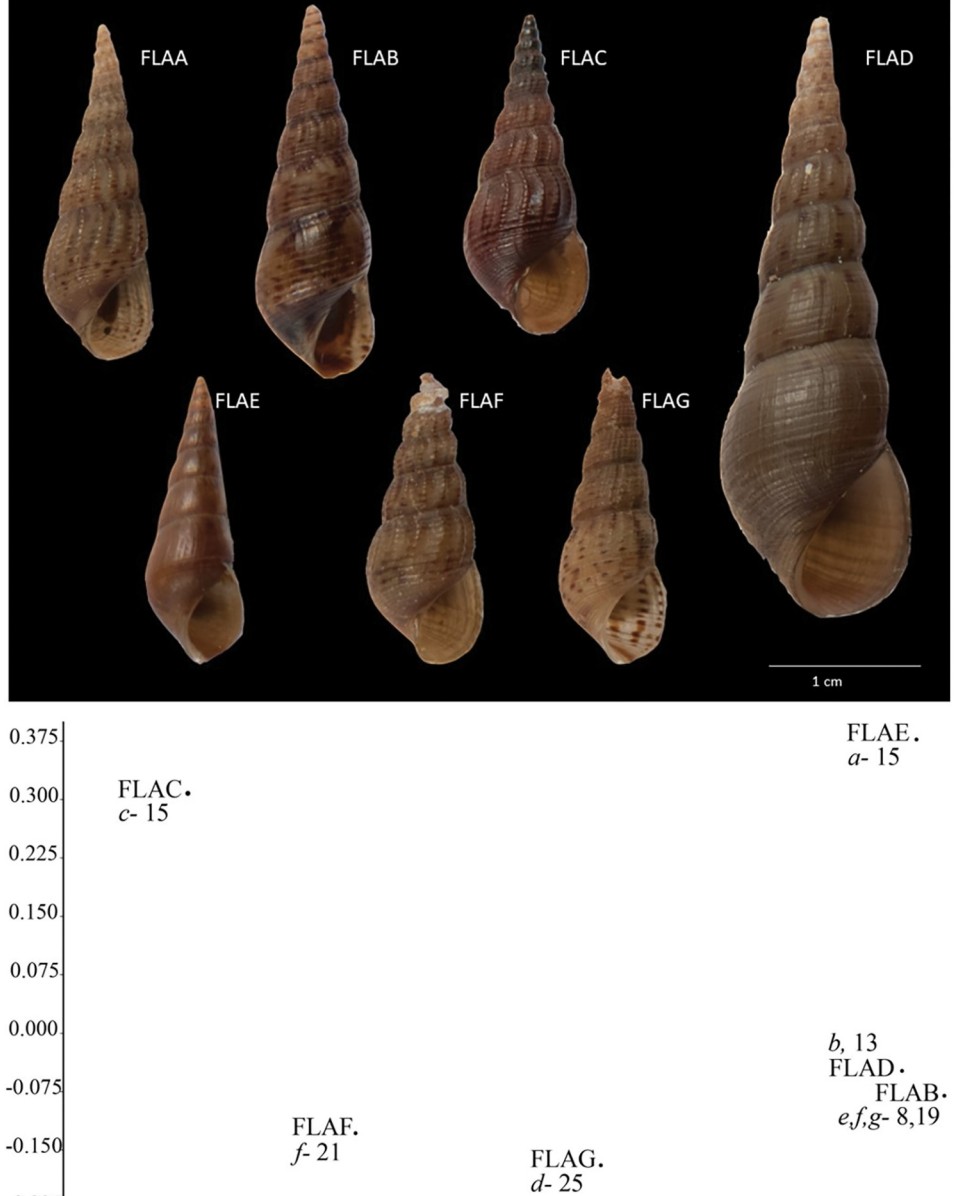

**Fig 3. (Upper) Images of Florida shell morphs (FLAA–FLAG).** Scale bar = 1 cm. (Lower). Non-dimensional scaling analyses (NDS) of Florida morphs (FLAA–FLAG) Stress = 0.0602. Corresponding haplotypes (lowercase italicized) and site numbers are shown on plot.

dynamics in Florida (i.e., serial founder events from multiple, putative source populations, or limited introductions and subsequent range expansions). We suggest, based on the geographic and phylogenetic patterns, that serial introductions are likely the primary mechanism of invasion dynamics to date. However, the degrees of local dispersal and the potential of hybridization still needs further scrutiny.

## Genetic diversity and invasion patterns

We identified 8 haplotypes with a high degree of genetic structuring without clear spatial patterns of distribution (Fig 1), which is similar to findings from Samadi et al. [31] and Facon et al. [24] from other world regions. These findings point away from naturally mediated range dispersal as the structuring mechanism for snail demographics in Florida. Further, the fact that 7 of 8 FLORIDA haplotypes were identical to WORLD haplotypes supports the hypothesis that several, independent invasions have occurred across the state. While it is not possible to know past modes of transport for invading snails in Florida, the fidelity of FLORIDA-specific haplotypes to WORLD haplotypes suggests that time scales needed for divergence from source populations in the global dataset have not occurred. However, FLORIDA haplotype *b* was unique and may represent a localized hybridization event at a site where these snails were first reported in 1971 [53].

Other studies have found that invaded regions have a high degree of haplotype overlap with native regions and provide estimates of invasion events. For instance, Facon et al. [24] showed at least six invasion events in South and Central America, while Harding et al. [30] suggested multiple invasions have occurred in central Texas. These studies and our current results demonstrate that multiple invasions are common for freshwater snails and provide evidence that propagule pressure is a driving factor for these successful invasions [54, 55].

In addition to haplotype evidence, historical records of *M. tuberculata* in Florida are useful for understanding observed distribution patterns (Table 1). Over a 5-year collection period (1966–1971), snails were found through much of the central and southern regions of the state, suggesting these snail populations were established years prior to collection dates. When our sites were compared against previous collection records, we found that Sites 13 and 15 date back to the early 1970s [53] while the first collections from our Site 4 were made in 2003 by Chadwick et al. [56]. Although populations may have established earlier than the late 1960s, it is difficult to determine if these snails are legacies of original invaders (e.g., parthenogenetic daughters) or from subsequent invasions. We know from our past research [35, 56, 57] that snails have been eliminated where past records of invasion exist. The extirpation of introduced populations is highly likely given that these snails are susceptible to mortality when water temperatures are sustained for several days below 15˚C [6], which can occur in southern Florida [58]. Therefore, finding multiple haplotypes in some locations and extirpations from other localities further supports serial invasions rather than local-scale range expansions.

All snails collected in Florida from our study were nested within clades containing sequences previously recognized as *M. tuberculata*, and the topology allowed us to identify likely source populations. Both clades included haplotypes from Malaysia that are also implicated in East African invasions [24, 59]. This is consistent with Chiu et al. [29] who suggested that haplotypes recovered in other studies of invaded regions (Caribbean, Central America, South America, Africa, and Miami, Florida, West Africa) mainly originated from Malaysia. Our results generally corroborate these phylogeographic patterns and provide additional evidence for serial invasions from multiple regions of origin. It is worth noting that these results are based on a single locus genetic marker and may underestimate the genetic diversity of these invasive snails [60].

## Shell morphology

We found that the use of shell form is ineffective for tracking new invasions in Florida given the lack of shell fidelity to specific haplotypes. Previous studies of the relationship between shell form and genetic diversity are mixed and depend on genetic markers and region of invasion. Samadi et al. (1999) [31] showed that shell forms aligned with invasive haplotypes in

invaded regions of Central and South America, and Facon et al. [24] suggested that shell morphs are correlated with genetic lineages. However, other studies showed weak links between shell forms and haplotype diversity. For instance, Duggan and Knox [16] point out in their study in New Zealand that shell characteristics were not reliable for identifying lineage origins, and overlapping shell morphologies between native and invasive *M. tuberculata* in African rift lakes were evidence of "camouflaged invasions" in the region [48]. In addition, relying on shell morphology led to the misidentification of *M. tuberculata* with *Stenomelania s. l.* [1], such that separation between these groups required habitat information (*Stenomelania* from marine/brackish waters and *M. tuberculata* from freshwaters) and confirmation of reproductive strategies as *Stenomelania s.l.* produce veliger larvae and *M. tuberculata* produce shelled juveniles in a brood pouch [1, 51]. We dissected individuals from populations in our study and found fully formed shelled individuals within brood pouches which supports that the taxa in our study are not currently recognized species of *Stenomelania*. Distribution records are further complicated by the lack of voucher specimens in some locality databases, while museum specimens often lack soft tissues needed to confirm the presence of brood pouches. Therefore, the invasion of Thiarids in Florida and elsewhere will require a taxonomic revision of this family to serve as a framework for snail identification. Additional studies on environmental limits, life histories, and updated records are needed to understand current and future invasive ranges of these species.

## Conclusions

Our results suggest that independent, serial introductions best explain the distribution of the invasive *M. tuberculata* species complex in Florida. However, shell morphology was not found to be a useful proxy for haplotype diversity; thus, shell morphology cannot be used to reconstruct invasion history of *M. tuberculata s.l.*, although it provides valuable information about phenotypic plasticity in this species complex. Our investigation further highlights the need for a careful taxonomic revision of *M. tuberculata s.l.* and related Thiarid species complexes. Finally, this data provides baseline information about the ongoing invasive species crisis in Florida which will hopefully help to inform management decisions moving forward, including efforts to minimize future invasions and recognize additional invasions of thiarid snails.

## Acknowledgments

Jessica Wooten and Nicholas (Reed) Alexander, Piedmont College, Georgia performed the DNA extraction and DNA amplification.

## Author Contributions

**Conceptualization:** Lori R. Tolley-Jordan, Michael A. Chadwick.

**Data curation:** Lori R. Tolley-Jordan.

**Formal analysis:** Lori R. Tolley-Jordan, Michael A. Chadwick, Jimmy K. Triplett.

**Funding acquisition:** Lori R. Tolley-Jordan.

**Methodology:** Lori R. Tolley-Jordan, Michael A. Chadwick, Jimmy K. Triplett.

**Writing – original draft:** Lori R. Tolley-Jordan.

**Writing – review & editing:** Lori R. Tolley-Jordan, Michael A. Chadwick, Jimmy K. Triplett.

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
