## [Decision Letter · Decision Letter 0]

29 Nov 2022

PONE-D-22-27150DNA barcoding indicates multiple invasions of the freshwater snail Melanoides tuberculata in Florida.PLOS ONE

Dear Dr. Chadwick,

Thank you for submitting your manuscript to PLOS ONE. After careful consideration, we feel that it has merit but does not fully meet PLOS ONE’s publication criteria as it currently stands. Therefore, we invite you to submit a revised version of the manuscript that addresses the points raised during the review process.

We look forward to receiving your revised manuscript.

Kind regards,

Hudson Alves Pinto, Ph.D

Academic Editor

PLOS ONE

Journal Requirements:

 "Funding was provided by United States Fish and Wildlife Service, Gulf States Marine Fisheries Division (project ID: FWS-800-037-2016-JSU) and the McCluer Scholar Award given to L. Tolley-Jordan by the School of Science, Jacksonville State University"

Additional Editor Comments:

The Reviewers presented several pertinent comments for this MS. I encourage the authors to revise the MS. I consider the possible occurrence of other thiarids (*Semisulcospira*) in the samples evaluated, as commented by Reviewer#3, should be ruled out. For this, I agree that a more in-depth analysis of the generated sequences, including comparative divergence data, needs to be included in the corrected version of the MS.

Reviewers' comments:

Reviewer's Responses to Questions

**Comments to the Author**

1. Is the manuscript technically sound, and do the data support the conclusions?

Reviewer #1: Yes

Reviewer #2: Partly

Reviewer #3: No

2. Has the statistical analysis been performed appropriately and rigorously? 

Reviewer #1: Yes

Reviewer #2: Yes

Reviewer #3: No

3. Have the authors made all data underlying the findings in their manuscript fully available?

Reviewer #1: Yes

Reviewer #2: Yes

Reviewer #3: Yes

4. Is the manuscript presented in an intelligible fashion and written in standard English?

Reviewer #1: Yes

Reviewer #2: Yes

Reviewer #3: Yes

5. Review Comments to the Author

Reviewer #1: After reviewing the manuscript titled “DNA barcoding indicates multiple invasions of the freshwater snail Melanoides tuberculata in Florida”, the authors have investigated the patterns of dispersal of snails in Florida through phylogenetic reconstruction including sequences from globally distributed snails using mitochondrial COI markers. This report covers important new ground for resolving the introduction of historical phylogeography and population genetic structure unique to Florida freshwater snails in relation to the world. As such it warrants publication in PLOS ONE as an important regional reference for sustained management and protection of evolutionary biodiversity in an area of progressive invasive disturbance due to anthropogenic impacts. The manuscript would benefit by integrating the different categorical results across biomarkers into a summary Figure 3 that illustrates the main introduction historical scenario described near the end of the Discussion and proposed by the authors as reflected by the title of the study. This should be mapped in historical detail onto the geographic region of Florida vs. the world and not just be summarized as an abstract conceptual diagram as in Fig. 1. Although this paper is well written, it wears some mistakes in data analysis and interpretation. The abstract is dreadful and looks as though it has been written by an inexperienced team member, without scrutiny by the rest of the team. The abstract will need to be rewritten to achieve clarity and give the reader the confidence that the manuscript is going to be worth reading. The described aims are not clear, and any hypotheses were proposed. In the introduction section, the author must have revised and described the geology of Florida in detail. It will help the readers understand the dispersed primarily through independent invasions, range expansion of populations through connected waterways, or a combination of both. In the Methodology section, there are some mistakes in the phylogenetic analysis. We suggested that the most appropriate model was used by SMS: Smart Model Selection in PhyML. We strongly suggested that the author should reconstruct the network topology that has generated a given invasive scenario. Their results need to be discussed in perspective. Despite the comprehensive data presented the results and discussion were not profoundly discussed. Figure 1, A map with more illustrations is needed for the paper. The current map only has the sampling locations’ initials on it. It is really difficult for someone not familiar with the river system in the region of Florida to understand this paper and get the main idea of the paper. While reading the manuscript we encountered several issues that need the most careful attention from the authors. Some issues are semantic/conceptual others are technical. We tried making my review as constructive as possible and hoped the authors find it useful. In my opinion, this manuscript does not meet the criteria for publication and must therefore be minor revised.

I have no minor comments.

Reviewer #2: Dear Authors:

I have reviewed the work, which is well structured and the effort in the collections is valued. It mainly has a molecular approach, with implications in the shape of the shell. The results obtained are similar to those proposed by other authors, corroborating the hypotheses raised, different income of the invasive species, and the non-correlation between the shape of the shell and the different haplotypes.

In the text I have left minor comments, perhaps the most important ones refer to the mix of sections. There is information in results that should be in Methodology. The same with table 5, which is mentioned for the first time in the Discussion (it should be in results, or write in another way so as not to cite a new table in that section). In the discussion, do not name figures and tables.

However, I do not find any innovation or discussion in the work that enhances its reading. It is based on a descriptive analysis of the phylogenetic tree (based on a single gene, as the title mentions), indicating where the populations could come from, but it does not go beyond that. Populations can be over 70 years old in the state of Florida, and there is possibly hybridization, hence the different haplotypes. Furthermore, the records in the areas of origin may have been reintroductions from other areas, as several authors have mentioned. See the possibility of analyzing sequences (of COI or another gene, 12S or 16S) from other parts of the world, since, in South and Central America, this snail is also scattered, and the origin could easily come from those places ( Florida has a large Latino population), regardless of the original populations. The specimens present in Florida could also come from populations present in the US (there is a 2019 study on the state of Texas), however, there are no other samples analyzed from that country. This information is interesting to know the invasion routes and thus have more solid information to elucidate the origins and the genetic flow.

On the other hand, no analysis of the vectors has been performed. The authors conclude that aquarism would be the possible way (just as other authors suggest, and surely it is that), without information to support it. Aquariums in the area should be studied, see what the traffic/trade is with the areas of origin, see the possible ports of entry.

In my opinion, there should be something more forceful in the discussion.

Reviewer #3: The manuscript describes a study that uses a single locus barcoding approach to identify and determine the origins of non-native thiarids, presumed to be primarily Melanoides tuberculata, in Florida. While the overall manuscript is well written and clear, and the figures nice. It starts from a false premise, and that is that all the thiarids other than Tarebia granifera, collected in Florida belong to a single taxon. Although the authors do not provide genetic distances or phylogenetic trees with meaningful branch lengths, it is clear from the shell figures and analyses that there is something other than simple deep divergences of populations going on here. Using the data uploaded by the authors, and a broader range of thiarid sequences available publicly, produces a dramatically different conclusion from that presented by the authors. As the authors present, there are multiple haplotypes from Florida that fall into distinct clades, but not all individuals in these clades are identified as M. tuberculata. At least two belong to the genus Stenomelania. While it is not possible to revise the taxonomy of the problematic thiarids within the scope of this study, it is possible for the authors to more robustly interpret their data within the framework of such taxonomic confusion. While it is clear there are multiple introductions, they are not of the same species. Additionally, these data are insufficient for delineating the source populations, simply because the geographic scope of sampling is not broad enough to make such determinations.

I suggest the authors take their data, which is well done and of broad interest, and place t within the appropriate taxonomic framework. This will allow the to clearly articulate the significance of multiple invaders in Florida, instead of continuing to confuse the situation by lumping these taxa into a single, non-monophyletic group.

One final suggestion, if the authors were to do a bit deeper sequencing, (more than a handful of individuals from each site) they are likely to find additional sympatric taxa with a wider distribution.

6. PLOS authors have the option to publish the peer review history of their article (what does this mean?). If published, this will include your full peer review and any attached files.

Reviewer #1: No

Reviewer #2: No

Reviewer #3: **Yes: **Kenneth A Hayes

---

## [Author Response · Author response to Decision Letter 0]

15 May 2023

We have addressed all comments and documents these in the attached documents.

---

## [Decision Letter · Decision Letter 1]

21 Jun 2023

PONE-D-22-27150R1DNA barcoding indicates multiple invasions of the freshwater snail Melanoides tuberculata in Florida.PLOS ONE

Dear Dr. Chadwick,

Thank you for submitting your manuscript to PLOS ONE. After careful consideration, we feel that it has merit but does not fully meet PLOS ONE’s publication criteria as it currently stands. Therefore, we invite you to submit a revised version of the manuscript that addresses the points raised during the review process.

We look forward to receiving your revised manuscript.

Kind regards,

Hudson Alves Pinto, Ph.D

Academic Editor

PLOS ONE

Additional Editor Comments:

I congratulate the authors for their efforts in the revision. However, I agree with the comments presented by Reviewer 3 on a more in-depth interpretation of phylogenetic analysis and taxonomic implications. I am convinced that other thiarid species than M. tuberculata are being misidentified in the Americas, and the present study needs to advance in this issue.

Reviewers' comments:

Reviewer's Responses to Questions

**Comments to the Author**

1. If the authors have adequately addressed your comments raised in a previous round of review and you feel that this manuscript is now acceptable for publication, you may indicate that here to bypass the “Comments to the Author” section, enter your conflict of interest statement in the “Confidential to Editor” section, and submit your "Accept" recommendation.

Reviewer #1: All comments have been addressed

Reviewer #3: All comments have been addressed

2. Is the manuscript technically sound, and do the data support the conclusions?

Reviewer #1: Yes

Reviewer #3: Partly

3. Has the statistical analysis been performed appropriately and rigorously? 

Reviewer #1: Yes

Reviewer #3: No

4. Have the authors made all data underlying the findings in their manuscript fully available?

Reviewer #1: Yes

Reviewer #3: Yes

5. Is the manuscript presented in an intelligible fashion and written in standard English?

Reviewer #1: Yes

Reviewer #3: Yes

6. Review Comments to the Author

Reviewer #1: After reviewing the manuscript titled " DNA barcoding indicates multiple invasions of the freshwater snail Melanoides tuberculata in Florida", the authors have investigated the patterns of dispersal of snails in Florida through phylogenetic reconstruction including sequences from globally distributed snails using mitochondrial COI markers. I feel that the manuscript has been significantly improved and satisfied with previous revisions. The whole manuscript and abstract are well-written and organized; hypothesis-test is obviously well defined. The data and analysis generally appear to be sound, the results are clear and interesting. After careful consideration, I feel that this manuscript will likely be suitable for publication.

Reviewer #3: While I applaud the authors for making substantial revisions to the manuscript, I was disappointed that the entire narrative is still focused on the premise that this is a single taxon introduced multiple times instead of the pattern shown of multiple species introduced. It is clear that both Stenomelania, Melanoides, several other genera in the Thiaridae are in critical need of revision. And while it is not the place for these authors to undertake those revisions in this study, it is their obligation to place their work and interpret in the objective and accurate context based on this fundamental understanding. The title still reflects this focus, and the discussion of the analyses are still centered around this false premise. It would be much more informative if the authors more clearly revised the title, introduction, analyses, and interpretation to reflect the reality of confused taxonomy, multiple species introductions, and emphasized the work needed to resolve this. The discussion could be improved by reflecting on the implications of the confused taxonomy. How can anyone be certain of what parasites are using what snails if one or both are not accurately and consistently identified. Historically parasitologists have identified the parasites hosted by various thiarids without any clear taxonomic understanding. This has lead to a long history of parasitological literature, and malacological literature rife with misinformation. Adding to that confusion without making a concerted effort to help identify a way forward and how we can better inform these relationships and identities is not something we should be publishing.

That said, I do think that the authors have data that can be used to inform both the taxonomic confusion, and other misinformation surrounding these highly invasive species. However, they need to do a better job of including broader taxonomic sampling in their analysis. While they have gone back and included additional taxa, they still cherry-picked taxa to include, which dramatically skews the topology of their tree, and the subsequent interpretations. Additionally, they still implicate the pet-trade as the culprit, but provide no data from surveys of aquarium shops or any other data that would support this. I have no doubts that the aquarium trade is responsible in some part, but the authors need to still provide data to support this conclusion.

Additionally, their justification for an outgroup is founded on a misunderstanding of what should be represented by an outgroup, and merely selecting an outgroup from another study, with very different aims, and more importantly different ingroup taxa is no a sound scientific approach. An outgroup should be chosen based on the question/hypotheses being addressed by the phylogenetic approach, and by the ingroup taxa of interest. In one section the authors state the used Tarebia granifera as the outgroup (pg 6 line 109), then later claim they selected Paludomus siamensis based on Wiggering et al. (2019), but they misspell the species on line 153 of page 8. More problematically, Wiggering et al. (2019) was addressing a fundamentally different question, about taxa in Thailand, and this outgroup is inappropriate for this analysis. Neither Tarebia or Paludomus are appropriate for outgroups in this situation, and the authors, if they examine Wiggering 2020 closely, they will see this. Thiara and additional Stenomelania species, and additional representatives are clearly needed to more fully understand the placement of taxa in Florida, and their affinities with other species across the globe.

As the authors point out, Wiggering 2020 clearly shows that Melanoides is polyphyletic, but so is Thiara, Stenomelania, and others, but the authors have not discussed the implications of this on their interpretation of multiple introductions, or on their shell analysis.

This really needs a major revision, not just adding comments reflecting this issues, but an actual interpretation and discussion that features the implications of the taxonomic problems and what it means for understanding introduction pathways.

7. PLOS authors have the option to publish the peer review history of their article (what does this mean?). If published, this will include your full peer review and any attached files.

Reviewer #1: No

Reviewer #3: No

---

## [Author Response · Author response to Decision Letter 1]

5 Sep 2023

Response to the editor’s general comments:

New Title: “DNA barcoding indicates multiple invasions of the freshwater snail Melanoides tuberculata sensu lato in Florida”

We agree that the previous draft of our manuscript did not emphasize the likelihood that multiple species of Thiarids currently identified as Melanoides in GenBank have invaded Florida. As such, we addressed this topic in two ways. First, we changed Melanoides tuberculata to M. tuberculata sensu lato in the revised title and throughout the manuscript. The use of sensu lato is the correct epithet to show that these individuals are potentially phylogenetically polyphyletic and morphologically cryptic (despite the fact that this paper is not intended to focus on this aspect of these invasive snails). Regardless of this tricky taxonomy, our results showed multiple lineages of M. tuberculata s.l. have invaded Florida, strongly supporting our contention of multiple invasions rather than a single invasion with subsequent dispersal. 

With respect to lack of detection of other species of invasive thiarids, we agree that this is problematic. However, our genetic data confirmed only the invasion of M. tuberculata s.l. and Tarebia granifera based on available GenBank information. We emphasize in the text the need for voucher specimens to be reported with locality records so that specimen identifications can be confirmed by taxonomic experts. Also, the prolific nature of the aquarium trade and online availability of additional Thiarids suggests that other genera are in the state and granular surveys are needed to detect new species.

Comments below provide detailed responses to reviewer 3.

Responses to reviewer 3 general comments:

First off, we very much appreciate the extensive time and effort of the reviewer in ensuring that clear, pointed comments on the extensive taxonomic issues of Thiarids. We have carefully addressed these comments in this revision (see below and in the track-changes version of the paper. Our previous version did not focus on the significance of taxonomic issues and invasion pathways of Melanoides tuberculata. As such, the previous revision did not highlight important components of documenting invasions such as voucher specimens, understanding life history characteristics, in addition to informative molecular markers. Our work clearly needed to be improved to emphasize that detecting many invasion events are difficult due to the cryptic nature of these snails and the need for taxonomic revision of this group. In addition, the likelihood that other species of Thiarids are in Florida but have been incorrectly identified (due to problematic taxonomy and shell phenotypic plasticity) are highlighted as confounding issues in understanding invasion dynamics. We stress that the inclusion of voucher specimens is critical for individuals that appear to be “M. tuberculata” given the high degree of phenotypic plasticity that masks invasions of other Thiarids, such as Stenomelania s.l. These changes reflect the complexity of investigating the invasion dynamics of cryptic snail species and we feel satisfactorily address concerns of Reviewer 3. 

This new version highlights these issues with respect to invasion dynamics throughout the text as extensive changes to the abstract, introduction, and discussion were made. In this updated version, the epithet Melanoides tuberculata was changed to Melanoides tuberculata sensu lato to reflect the taxonomic issues of this species complex. Further, we included additional citations by Strong et al. 2011 that showed Paludomidae is a sister clade to Thiaridae to justify our choice of outgroup. 

Specific answers to the comments and concerns of reviewer 3:

Reviewer comments: I applaud the authors for making substantial revisions.

Response: Thank you for the kind response of the considerable efforts we made in the last revision, and we feel that the paper is much improved. We appreciate the attention to detail of all reviewers and editors to improve this manuscript.

Reviewer comment: Narrative is still focused on the premise that this is a single taxon introduced multiple times instead of the pattern shown of multiple species introduced

Response: We agree that we missed the mark on clearly stating that this species is polyphyletic. As such, sensu lato appears in the title, the first sentence of the abstract, the beginning of the paragraph in the introduction. In the abstract and discussion, we refer to M. tuberculata as a species complex. We highlight the placement of Florida haplotypes into two clades that resulted in the invasion of a species complex into Florida- not a single species.

Reviewer comment: It is clear that both Stenomelania, Melanoides, several other genera in the Thiaridae are in critical need of revision. And while it is not the place for these authors to undertake those revisions in this study, it is their obligation to place their work and interpret in the objective and accurate context based on this fundamental understanding

Response: We recognize that the significance of the unresolved Thiaridae taxonomy was not emphasized in the previous drafts and took great care to highlight these issues throughout the text in this revision.

Reviewer comment: The title still reflects this focus, and the discussion of the analyses are still centered around this false premise.

Response: We have updated the title to include the term sensu lato to show that M. tuberculata is a species complex that in need of taxonomic revision.

Reviewer comment: It would be much more informative if the authors more clearly revised the introduction, analyses, and interpretation to reflect the reality of confused taxonomy, multiple species introductions, and emphasized the work needed to resolve this.

Response: We agree and included much more detail on the significance of taxonomic uncertainty in describing invasion patterns.

Reviewer comment: The discussion could be improved by reflecting on the implications of the confused taxonomy

Response: Please see updated information in the title, abstract, introduction, methods, and discussion.

Reviewer comment: The discussion could be improved by reflecting on the implications of the confused taxonomy

Response: We appreciate that the reviewer has read previous works about the importance of Melanoides tuberculata s.l. as a host for many trematode parasites, we have not covered this issue in this manuscript. We are unclear as to how to address this comment. 

Reviewer comment: Additionally, they still implicate the pet-trade as the culprit, but provide no data from surveys of aquarium shops or any other data that would support this.

Response: We refer to other papers to support this assertion, but we suggest, multiple times, that further researcher on introduction mechanisms are needed. Our work highlights the need for more research on the significance of the aquarium trade.

Reviewer Comment: Justification for an outgroup is founded on a misunderstanding of what should be represented by an outgroup. In one section the authors state the used Tarebia granifera as the outgroup (pg 6 line 109), then later claim they selected Paludomus siamensis based on Wiggering et al. (2019), but they misspell the species on line 153 of page 8. More problematically, Wiggering et al. (2019) was addressing a fundamentally different question, about taxa in Thailand, and this outgroup is inappropriate for this analysis. Neither Tarebia or Paludomus are appropriate for outgroups in this situation, and the authors, if they examine Wiggering 2020 closely, they will see this. Thiara and additional Stenomelania species, and additional representatives are clearly needed to more fully understand the placement of taxa in Florida, and their affinities with other species across the globe.

Response: We corrected the error about Tarebia as an outgroup which was an oversight in previous edits of the manuscript.. Outgroup selection was initially based on data from Wiggering 2020. Based on this comment, we included reference by Strong et al. 2011 that showed Paludomidae is sister to Thiaridae and The sequence from Wiggering 2020 was used due to the presence of a voucher specimen. We are unclear as why another outgroup(s) other than Paludomus siamensis is required given the fact that our phylogeny mirrors Wiggering 2020 and phylogenies of M. tuberculata generated in other studies.

Reviewer Comment: Wiggering 2020 clearly shows that Melanoides is polyphyletic, but so is Thiara, Stenomelania, and others, but the authors have not discussed the implications of this on their interpretation of multiple introductions, or on their shell analysis.

Response: In the discussion, we updated the section on Shell Morphology to reflect previous findings that phenotypic plasticity of shells within and among genera of Thiaridae is so pervasive that evidence of life histories and reproductive strategies are needed to parse taxa into different lineages. We included text highlighting the importance of voucher specimens, including soft tissues when available, are needed to identify specimens of Thiarids.

Reviewer Comment: This really needs a major revision, not just adding comments reflecting this issues, but an actual interpretation and discussion that features the implications of the taxonomic problems and what it means for understanding introduction pathways.

Response: We have markedly altered the text in this submission to reflect the concerns of the importance of taxonomy on documenting invasion patterns. We feel that this new version highlights the interactions of polyphyly, phenotypic plasticity, and overall taxonomic problems within and among lineages of Thiaridae complicate our understanding of invasion dynamics in this study. Further, without including references to taxonomic concerns an accurate reflection of the extent and severity of an invasion cannot be determined.

---

## [Editor Report · Decision Letter 2]

15 Sep 2023

DNA barcoding indicates multiple invasions of the freshwater snail Melanoides tuberculata sensu lato in Florida

PONE-D-22-27150R2

Dear Dr. Chadwick,

We’re pleased to inform you that your manuscript has been judged scientifically suitable for publication and will be formally accepted for publication once it meets all outstanding technical requirements.

Kind regards,

Hudson Alves Pinto, Ph.D

Academic Editor

PLOS ONE

Additional Editor Comments (optional):

I congrats the authors for the high-quality discussion. I agree use of "sensu lato" is currently a good option for this complex taxonomic scenary. I thank the authors for this very relevant contribution.
---

## [Editor Report · Acceptance letter]

25 Sep 2023

PONE-D-22-27150R2 

DNA barcoding indicates multiple invasions of the freshwater snail *Melanoides tuberculata sensu lato* in Florida. 

Dear Dr. Chadwick:

I'm pleased to inform you that your manuscript has been deemed suitable for publication in PLOS ONE. Congratulations! Your manuscript is now with our production department. 

Kind regards, 

on behalf of

Dr. Hudson Alves Pinto 

Academic Editor

PLOS ONE